# Retrotransposon-mediated disruption of a chitin synthase gene confers insect resistance to *Bacillus thuringiensis* Vip3Aa toxin

Zhenxing Liu[1]☯, Chongyu Liao[1]☯, Luming Zou[1]☯, Minghui Jin[1]☯, Yinxue Shan[1]☯, Yudong Quan[2], Hui Yao[1], Lei Zhang[1], Peng Wang[1], Zhuangzhuang Liu[1], Na Wang[1], Anjing Li[3], Kaiyu Liu[3], Bruce E. Tabashnik[4], David G. Heckel [1,5]*, Kongming Wu[2]*, Yutao Xiao [1]*

**1** Shenzhen Branch, Guangdong Laboratory of Lingnan Modern Agriculture, Key Laboratory of Gene Editing Technologies, Ministry of Agriculture and Rural Affairs, Agricultural Genomics Institute at Shenzhen, Chinese Academy of Agricultural Sciences, Shenzhen, China, **2** The State Key Laboratory for Biology of Plant Disease and Insect Pests, Institute of Plant Protection, Chinese Academy of Agricultural Sciences, West Yuanmingyuan Road, Beijing, China, **3** Institute of Entomology, School of Life Sciences, Central China Normal University, Wuhan, China, **4** Department of Entomology, University of Arizona, Tucson, Arizona, United States of America, **5** Department of Entomology, Max Planck Institute for Chemical Ecology, Jena, Germany

☯ These authors contributed equally to this work.
\* heckel@ice.mpg.de (DGH); wukongming@caas.cn (KW); xiaoyutao@caas.cn (YX)

**Data Availability Statement:** All data are fully available without restriction. All relevant data are within the manuscript and its Supporting Information files. Sequences of the chitin

## Abstract

The vegetative insecticidal protein Vip3Aa from *Bacillus thuringiensis* (Bt) has been produced by transgenic crops to counter pest resistance to the widely used crystalline (Cry) insecticidal proteins from Bt. To proactively manage pest resistance, there is an urgent need to better understand the genetic basis of resistance to Vip3Aa, which has been largely unknown. We discovered that retrotransposon-mediated alternative splicing of a midgut-specific chitin synthase gene was associated with 5,560-fold resistance to Vip3Aa in a laboratory-selected strain of the fall armyworm, a globally important crop pest. The same mutation in this gene was also detected in a field population. Knockout of this gene via CRISPR/Cas9 caused high levels of resistance to Vip3Aa in fall armyworm and 2 other lepidopteran pests. The insights provided by these results could help to advance monitoring and management of pest resistance to Vip3Aa.

## Introduction

Insecticidal proteins from the bacterium *Bacillus thuringiensis* (Bt) have been used widely in pest control for nearly a century and have protected transgenic crops from insect damage since 1996 [1]. In addition to direct protection from feeding damage, benefits of transgenic Bt crops have included regional suppression or even eradication of pest populations, reduced damage to non-transgenic crops, reduced use of chemical insecticides, and compatibility with biological control by natural enemies [2–5]. However, planting of transgenic Bt crops including cotton, maize, and soybean on a cumulative total of more than 1.5 billion hectares since

synthases and the LTR retrotransposon insertion have been deposited in GenBank (Accession Numbers OR669300–OR669302).

**Funding:** This work was supported by grants from Sci-Tech Innovation 2030 Agenda (2022ZD04021) to YX; National Natural Science Foundation of China (32372546) to YX and (32001944) to MJ; Innovation Program of Chinese Academy of Agricultural Science (CAAS-CSCB-202303) to MJ; Agricultural Science and Technology Innovation Program of Chinese Academy of Agricultural Sciences to YX; Major Projects of Basic Research of Science, Technology and Innovation Commission of Shenzhen Municipality to YX; Senior Talents Project of Guangdong (2021A1313030029) to YX; and the Max-Planck-Gesellschaft to DGH. The funders had no role in study design, data collection and analysis, decision to publish, or preparation of the manuscript.

**Competing interests:** The authors have declared that no competing interests exist

**Abbreviations:** AP, aspartate protease; BSA, bulked segregant analysis; BWA, Burrows–Wheeler aligner; CCS, circular consensus sequence; CI, confidence interval; LTR, long terminal repeat; RR, resistance ratio; RT, reverse transcriptase; SDS-PAGE, sodium dodecyl sulfate-polyacrylamide gel electrophoresis; SNP, single nucleotide polymorphism; SS, susceptible strain; TSD, target site duplication; WGS, whole-genome sequencing.

1996 has selected intensely for pest adaptation [6]. At least 26 cases of practical resistance that decrease the efficacy of transgenic crops producing crystalline (Cry) insecticidal proteins from Bt have been documented in some populations of 11 pest species in 7 countries [6].

To counter resistance to Cry proteins, farmers have planted Bt crops that produce the vegetative insecticidal protein Vip3Aa simultaneously with one or more Cry proteins [7–9]. In the bacteria, Vip proteins are produced in the vegetative stage [10]. Vip3Aa is especially useful because of its efficacy against pests that are resistant to Cry proteins, which reflects its distinct mode of action [10–15]. Moreover, practical resistance to Vip3Aa in the field has not been reported yet. However, laboratory-selected resistance to Vip3A has been documented in several pests [12,13,16–21] and early warnings of field-evolved resistance to Vip3Aa have been reported for the major lepidopteran pests the corn earworm (*Helicoverpa zea*) and fall armyworm (*Spodoptera frugiperda*) [8,9,22]. Knowledge of the genetic basis of pest resistance to Bt proteins can be useful for monitoring and managing resistance in the field [23], but is lacking and urgently needed for Vip3Aa.

Knowledge of the genetic basis of resistance to Vip3Aa is especially important for the invasive pest *S. frugiperda*, which recently expanded its ancestral range in the Americas to include Africa, Asia, and Australia [24]. Its high fecundity and migratory tendencies pose serious problems for agriculture in the newly colonized regions [25]. Also, it has rapidly evolved practical resistance to Bt crops producing Cry proteins [9]. Previous studies have reported several proteins that are putative receptors for Vip3Aa or otherwise affect susceptibility to Vip3Aa [26–29]. However, we are not aware of evidence linking these proteins with resistance to Vip3Aa. Also, some aspects of inheritance of resistance to Vip3Aa such as dominance have been studied in laboratory-selected strains [12,13,17,18,20,30], but the specific genes involved have not been identified previously. Down-regulation of transcription factor *SfMyb* was associated with resistance to Vip3Aa in the laboratory-selected DH-R strain of *S. frugiperda*, but the results identified a set of candidate genes that might mediate the resistance rather than demonstrating which genes are actually involved [31]. Also, the DH-R strain had only 206-fold resistance to Vip3Aa [31], whereas some other strains had >1,000-fold resistance [12,20,32], for which no specific genetic basis has been reported previously as far as we know.

Here, we report that in *S. frugiperda*, naturally occurring disruption of a midgut-specific chitin synthase gene (*SfCHS2)* by a retrotransposon insertion was associated with 5,560-fold resistance to Vip3Aa. The resistance allele with the retrotransposon insertion that was identified from a laboratory-selected strain also was detected in the field. Furthermore, knockout of this gene via CRISPR/Cas9 in susceptible strains conferred high levels of resistance to Vip3Aa in *S. frugiperda* and 2 other major lepidopteran pests (*Spodoptera litura* and *Mythimna separata*). These results imply that the encoded chitin synthase protein plays an essential role in the mode of action of Vip3Aa against at least 3 major pests.

## Results

### Selection for resistance to Vip3Aa, survival on maize, and lack of strong cross-resistance

We established a Vip3Aa-resistant strain of *S. frugiperda* (Sfru_R3) from a susceptible strain (SS, previously called DH-S [31]) that originated from insects collected in 2019 from non-Bt maize in Ruili City, Dehong Prefecture of the Yunnan province in southwestern China. We selected Sfru_R3 for 17 generations on diet treated with gradually increasing concentrations of Vip3Aa (S1 Table), then compared strains based on the $EC_{50}$, which is the concentration (μg Vip3Aa per $cm^2$ diet) causing 50% of larvae to die or not reach third instar after 7 days. The

$EC_{50}$ of Vip3Aa was 428 for Sfru_R3 versus 0.077 for SS, which yields a resistance ratio (RR) of 5560 for Sfru_R3 (calculated as the $EC_{50}$ of Sfru_R3 divided by the $EC_{50}$ of SS, Table 1).

On transgenic maize plants producing Vip3Aa, survival was higher for Sfru_R3 (23%) than SS (0%, Fisher's exact test, $P = 10^{-16}$, S2 Table). On non-transgenic maize plants, survival also was higher for Sfru_R3 (84%) than SS (61%, Fisher's exact test, $P = 10^{-7}$, S2 Table). For Sfru_R3, survival on Vip3Aa maize relative to non-transgenic maize was 28% (S2 Table). Survival on transgenic maize producing Bt protein Cry1Ab did not differ significantly between Sfru_R3 (3.7%) and SS (1.4%, Fisher's exact test, $P = 0.22$, S2 Table). Results of diet bioassays show that selection for 5,560-fold resistance to Vip3Aa in Sfru_R3 did not cause strong cross-resistance to Cry1Ab (RR = 2.0), Cry1Ac (RR = 0.9), Cry1Fa (RR = 2.0), or Cry2Ab (RR = 1.8, S3 Table).

## Inheritance of Vip3Aa resistance

Inheritance of Vip3Aa resistance was autosomal, based on the similar $EC_{50}$ for the $F_1$ progeny from the 2 reciprocal crosses between Sfru_R3 and SS (0.27 and 0.26 µg Vip3Aa per $cm^2$ diet, Table 1). We also measured the dominance parameter $h$, which varies from 0 for completely recessive to 1 for completely dominant [33]. The dominance of resistance to Vip3Aa decreased as concentration increased, with $h = 0.61, 0.45, 0.065$, and 0 at concentrations of 0.25, 0.5, 1, and 2 µg Vip3Aa per $cm^2$ diet, respectively (S4 Table). In bioassays at a concentration of 2 µg Vip3Aa per $cm^2$ diet, observed mortality did not differ significantly from the mortality expected with resistance conferred by a single gene, either for progeny from backcrosses ($F_1 \times$ Sfru_R3, Fisher's exact test, $P = 0.20$; 53% observed versus 50% expected) or $F_2$ progeny ($F_1 \times$ $F_1$, Fisher's exact test, $P = 0.39$, 78% observed versus 75% expected, S5 Table).

## Identification of a candidate gene linked with resistance to Vip3Aa

To identify the genomic region associated with Vip3Aa resistance in Sfru_R3, we conducted a bulked segregant analysis (BSA) using genomic DNA from the parental moths from Sfru_R3 and SS and 2 groups of larvae from the $F_2$ progeny of the cross between strains (S1 Fig). After 5 days, $F_2$ larvae that developed to third instar on diet containing 4 µg Vip3Aa per $cm^2$ diet were scored as resistant ($F_2$-R, $n = 89$) and those that did not develop to third instar on diet containing 0.1 µg Vip3Aa per $cm^2$ diet were scored as susceptible ($F_2$-S, $n = 75$) (S1 Fig). The initial mapping identified a region from 6.56 to 10.92 Mb on chromosome 1 associated with resistance to Vip3Aa, based on high values for the ΔSNP index (Fig 1A and 1B).

**Table 1. Responses to Vip3Aa of a susceptible strain (SS), resistant strain (Sfru_R3), and progeny from crosses.**

| Strain or cross | $n$[a] | $EC_{50}$ (95% CI)[b] | Slope ± SE | RR[c] |
|---|---|---|---|---|
| SS | 144 | 0.077 (0.062–0.092) | 4.7 ± 0.9 | 1.0 |
| Sfru_R3 | 156 | 428 (312–617) | 4.4 ± 1.1 | 5,560 |
| $F_{1a}$: Sfru_R3♀ × SS♂ | 144 | 0.27 (0.23–0.32) | 5.9 ± 1.1 | 3.5 |
| $F_{1b}$: Sfru_R3♂ × SS♀ | 144 | 0.26 (0.14–0.55) | 3.8 ± 0.6 | 3.4 |
| $F_2$ ($F_1 \times F_1$)[d] | 2,304 | 0.33 (0.20–0.94) | 1.1 ± 0.1 | 4.3 |
| Backcross ($F_1 \times$ Sfru_R3)[d] | 4,608 | 2.97 (1.04–9.52) | 1.4 ± 0.1 | 38.6 |

[a] Number of neonates tested.

[b] Median effective concentration ($EC_{50}$); concentration that caused 50% of neonates to die or fail to advance to the third instar in 7 days and its 95% CI in µg Vip3Aa per $cm^2$ diet.

[c] Resistance ratio: $EC_{50}$ for a strain or progeny from a cross divided by the $EC_{50}$ for SS.

[d] Results pooled from all possible crosses for $F_2$ and $F_1 \times$ Sfru_R3 backcross progeny, respectively.

CI, confidence interval; RR, resistance ratio; SS, susceptible strain.

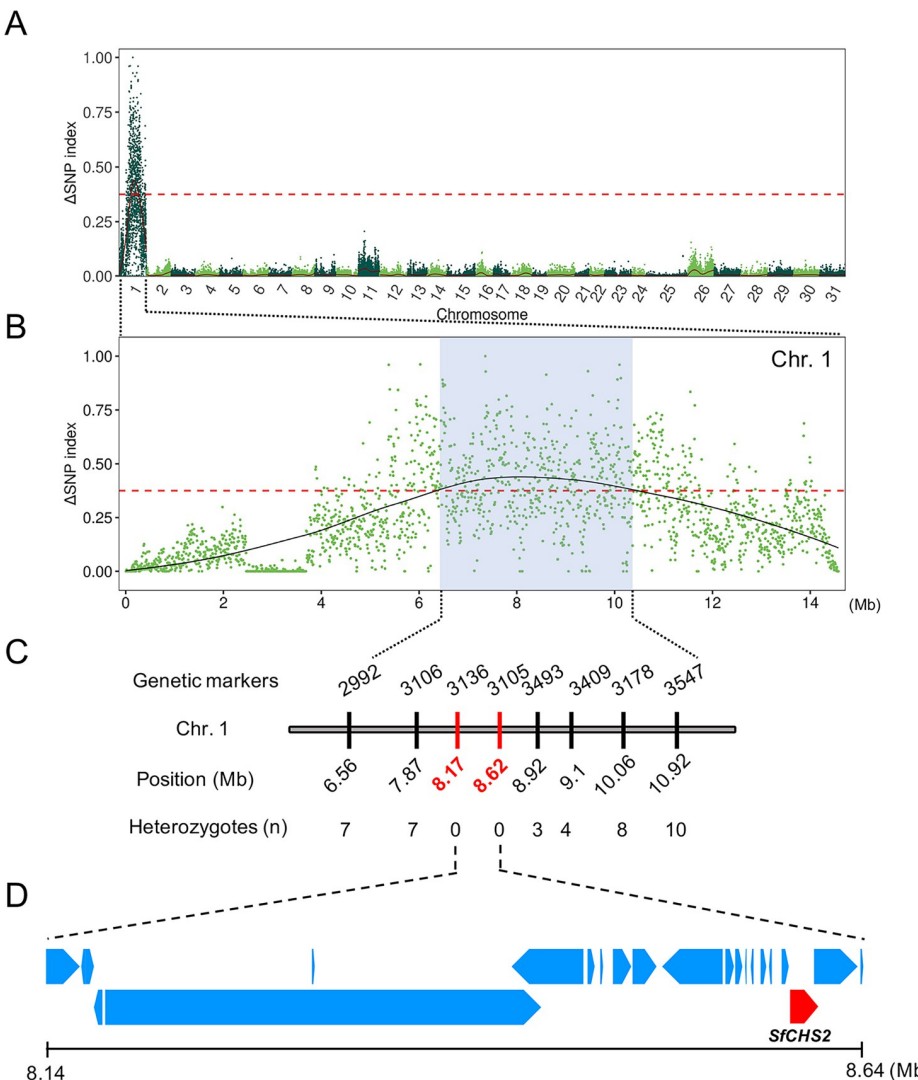

**Fig 1. Mapping Vip3Aa resistance in the *S. frugiperda* genome.** (A) Manhattan plot from BSA showing a ΔSNP index peak in chromosome 1 reflecting a high proportion of alleles associated with resistance to Vip3Aa. (B) Chromosome 1 with ΔSNP index highest from 6.56 to 10.92 Mb (blue shading). The solid line indicates the LOESS regression fitted ΔSNP index value. The dashed red line in (A) and (B) indicates the threshold for the top 1% of values for the ΔSNP index. (C) Fine-scale mapping shows complete linkage with markers at 8.17 and 8.62 Mb. Only the last 4 digits are shown for the 8 markers from 6.56 to 10.92 Mb. The full name for each marker starts with LOC11827. (D) Genes from 8.14 to 8.64 Mb including chitin synthase 2 (*SfCHS2*). The data underlying this figure can be found in S1 Data and https://doi.org/10.5281/zenodo.11395059. BSA, bulked segregant analysis; SNP, single nucleotide polymorphism.

In fine-scale mapping with 8 genetic markers from 6.56 to 10.92 Mb on chromosome 1, the markers at 8.17 and 8.62 Mb were tightly linked with resistance (Fig 1C). All $F_2$-R larvae tested were homozygous for the alleles derived from Sfru_R3. By contrast, for the other 6 genetic markers, heterozygotes were detected in $F_2$-R progeny and the percentage of heterozygotes increased with the distance from 8.17 and 8.62 Mb.

Of the 19 annotated genes from 8.17 to 8.64 Mb on chromosome 1, RNA-Seq of midguts of fifth instar larvae showed relatively high transcription of 3: 3105, 3137, and 3449 (S6 Table, all IDs start with LOC11827, only the last 4 digits are shown here). Structural analysis of the

predicted proteins encoded by the 19 genes showed that only 3105 and 3138 were expected to produce proteins associated with the plasma membrane. The value of FPKM (fragments per kilobase of transcript per million mapped reads) was 400 times higher for 3105 (111.97) than 3138 (0.28) (S6 Table). Taken together, the RNA-seq and predicted protein structure analyses identified 3105 as the primary candidate gene.

Locus 3105 encodes a chitin synthase (CHS), which is a membrane-integral glycosyl-transferase that catalyzes chitin biosynthesis by transferring GlcNAc from UDP-GlcNAc to a growing chitin chain [34,35]. Locus 3105 and adjacent 3150 are *SfCHS2* and *SfCHS1*, respectively (S2 Fig, previously also called *SfCHSB* and *SfCHSA*, respectively [36]). RNA-seq showed that FPKM in the larval midgut was 90 times higher for *SfCHS2* than *SfCHS1* (S6 Table), confirming our focus on *SfCHS2* as the primary candidate gene. The *SfCHS2* gene we sequenced contains an open reading frame of 4,572 bp encoding a protein of 1,524 amino acids with a predicted molecular mass of 174 kDa (GenBank OR669301). The SfCHS2 protein is a class B chitin synthase [36–38]. Class B chitin synthases occur in gut epithelial cells and produce chitin for the peritrophic matrix, which is important for diges-tion and protection from pathogens [34,35,39–42].

## A retrotransposon insertion associated with reduced abundance of wild-type *SfCHS2* transcripts

Results from Sanger sequencing of the full-length genomic sequences of *SfCHS2* from Sfru_R3 and SS amplified by PCR revealed a 6,427-bp insertion between exon 21 and intron 21 in Sfru_R3 (Fig 2A). BLAST results demonstrated that the insert is a long terminal repeat (LTR) retrotransposon, which we name Yaoer (幺蛾, which means wicked thing). Yaoer is composed of 2 LTRs (LTRa and LTRb) at the 5′ and 3′ terminal and a protein encoding sequence that contains the aspartate proteases (AP), reverse transcriptase (RT), ribonuclease H (RNase H), and integrase (INT) domains (Fig 2B). The insertion point was identified as a target site dupli-cation (TSD) of GAAGG immediately upstream of LTRa and downstream of LTRb (S3 Fig). The CHS2 protein encoded by the disrupted sequence containing Yaoer is expected to have a truncated C-terminal lumenal domain called C7 (S4 Fig).

Although Yaoer is inserted between exons 21 and 22 in Sfru_R3, the original 5′ splice site of intron 21 is intact and downstream from Yaoer (S3 Fig). Thus, Yaoer and intron 21 could be spliced out. To determine whether this occurs, we examined the transcripts of *SfCHS2* from Sfru_R3 using primers at exons 21 and 22. Results from gel electrophoresis and Sanger sequencing show that Sfru_R3 produced wild-type transcripts without Yaoer and mutant tran-scripts with Yaoer (S5 Fig). Mutant transcripts with intron 21 and Yaoer were also detected in Sfru_R3 (S5 Fig). Results from Iso-Seq of *SfCHS2* confirm that Sfru_R3 produced wild-type transcripts without Yaoer and mutant transcripts with Yaoer (S5 Fig). Total *SfCHS2* transcript abundance did not differ between SS ($1.0 \pm 0.2$) and Sfru_R3 ($1.8 \pm 0.3$, $t$ test, $t = 2.2$, df = 6, $P = 0.07$, Fig 2E). However, wild-type *SfCHS2* transcripts were 12 times more abundant in SS ($1.0 \pm 0.16$) than Sfru_R3 ($0.08 \pm 0.02$, $t$ test, $t = 5.6$, df = 6, $P = 0.001$, Fig 2H).

## Vip3Aa resistance genetically linked with Yaoer insertion and mutant *SfCHS2* transcripts

Analogous to our screening of $F_2$ larvae described above, we screened $F_7$ larvae from the cross between Sfru_R3 and SS to generate 2 groups: $F_7$-R (resistant) and $F_7$-S (susceptible). For this experiment, >10,000 $F_7$ neonates were exposed to diet containing 24 μg Vip3Aa per $cm^2$ and 2,400 larvae were exposed to diet containing 0.1 μg Vip3Aa per $cm^2$ diet. After 5 days, the 214 largest survivors from the diet with a high concentration of Vip3Aa formed the $F_7$-R group

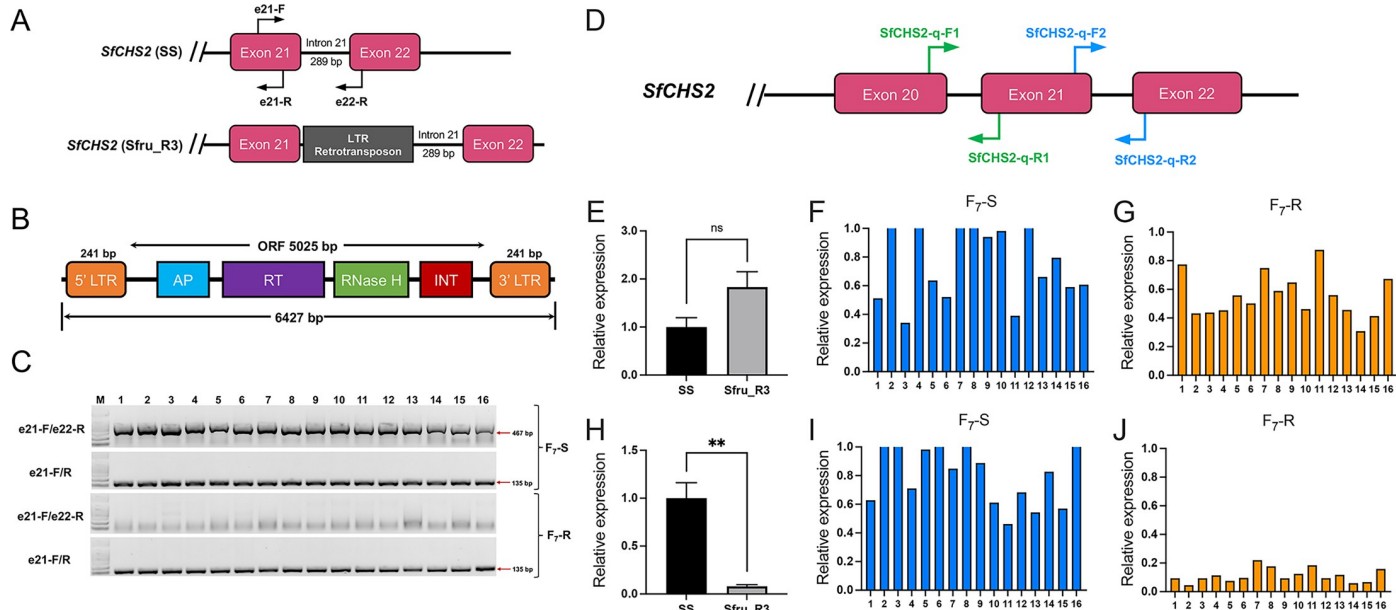

**Fig 2. Retrotransposon insertion associated with resistance to Vip3Aa and reduced abundance of wild-type *SfCHS2* transcripts.** (A) Exon 21 to 22 of *SfCHS2* in the SS (susceptible) and Sfru_R3 (Vip3Aa-resistant) strains showing the LTR retrotransposon (Yaoer) insertion. (B) Yaoer with LTR regions at the 5′ and 3′ termini and the intervening region encoding AP, RT, ribonuclease H (RNase H), and integrase (INT). (C) *SfCHS2* genotype of $F_7$ Vip3Aa-susceptible ($F_7$-S) and resistant ($F_7$-R) larvae determined using gDNA from individual larvae and primers shown in (A). The e21-F/e22-R primers flanking intron 21 produced a strong band indicating wild-type from all 16 $F_7$-S larvae and no $F_7$-R larvae. The positive control primers (e21-F/e21-R) in exon 21 produced a strong band in all $F_7$-S and $F_7$-R larvae. The molecular weight marker containing DNA with length of 100, 250, 500, 750, 1,000, and 2,000 bp was used for agarose gel electrophoresis analysis. (D) Primers for analyzing relative transcript abundance of *SfCHS2* via RT-qPCR. Primers SfCHS2-q-F1/R1 in exons 20 and 21 evaluated total transcript abundance of *SfCHS2* in SS versus Sfru_R3 (E) and individuals from $F_7$-S (F) versus $F_7$-R (G). Primers SfCHS2-q-F2/R2 flanking intron 21 evaluated wild-type transcript abundance of *SfCHS2* in SS versus Sfru_R3 (H) and individuals from $F_7$-S (I) versus $F_7$-R (J). Transcript abundance of *SfCHS2* in Sfru_R3 and individuals from $F_7$-S and $F_7$-R was normalized to the fold value of $2^{-\Delta Ct}$ relative to SS. Bars for SS and Sfu_R3 in (E) and (H) show mean relative transcript abundance ± SEM. Based on *t* tests, NS in (E) indicates no significant difference between strains in total transcript abundance and ** in (H) indicates SS greater than Sfu_R3 ($P < 0.01$, see text for details). The data underlying this figure can be found in S2 Data. AP, aspartate protease; LTR, long terminal repeat; RT, reverse transcriptase; SS, susceptible strain.

and the 109 smallest survivors from the diet with a low concentration of Vip3Aa diet formed the $F_7$-S group. Homozygosity for the Yaoer insertion occurred in all 214 $F_7$-R larvae and none of the 109 $F_7$-S larvae, demonstrating tight genetic linkage between this insertion and resistance to Vip3Aa (Fisher's exact test, $P = 10^{-88}$).

Results from RT-qPCR analysis of *SfCHS2* transcripts from $F_7$ larvae show strong genetic linkage between resistance and the percentage of mutant transcripts. For $F_7$-S larvae, mutant transcripts were absent as indicated by the slightly but not significantly higher abundance for wild-type transcripts (1.2 ± 0.2) than total transcripts (0.95 ± 0.14, *t* test, $t = 0.82$, df = 30, $P = 0.42$). Conversely, for $F_7$-R larvae, abundance was 5 times higher for total transcripts (0.56 ± 0.04) than wild-type transcripts (0.11 ± 0.01, *t* test, $t = 11$, df = 30, $P < 0.0001$), which indicates 80% of transcripts were mutant.

## CRISPR/Cas9-mediated knockout of *CHS2* confers resistance to Vip3Aa in 3 lepidopteran pests

Using CRISPR/Cas9, we discovered that knocking out the *CHS2* gene with a double sgRNA strategy conferred high levels of resistance to Vip3Aa in 3 strains of *S. frugiperda* (SfCHS2-KO-A, SfCHS2-KO-B, and SfCHS2-KO-C) and 1 strain each of the major lepidopteran pests *S. litura* (SlCHS2-KO) and *M. separata* (MsCHS2-KO) (Fig 3 and S7 Table). For SfCHS2-KO-A, we generated a 1,002-bp deletion between exons 4 and 5 (Fig 3A). The altered

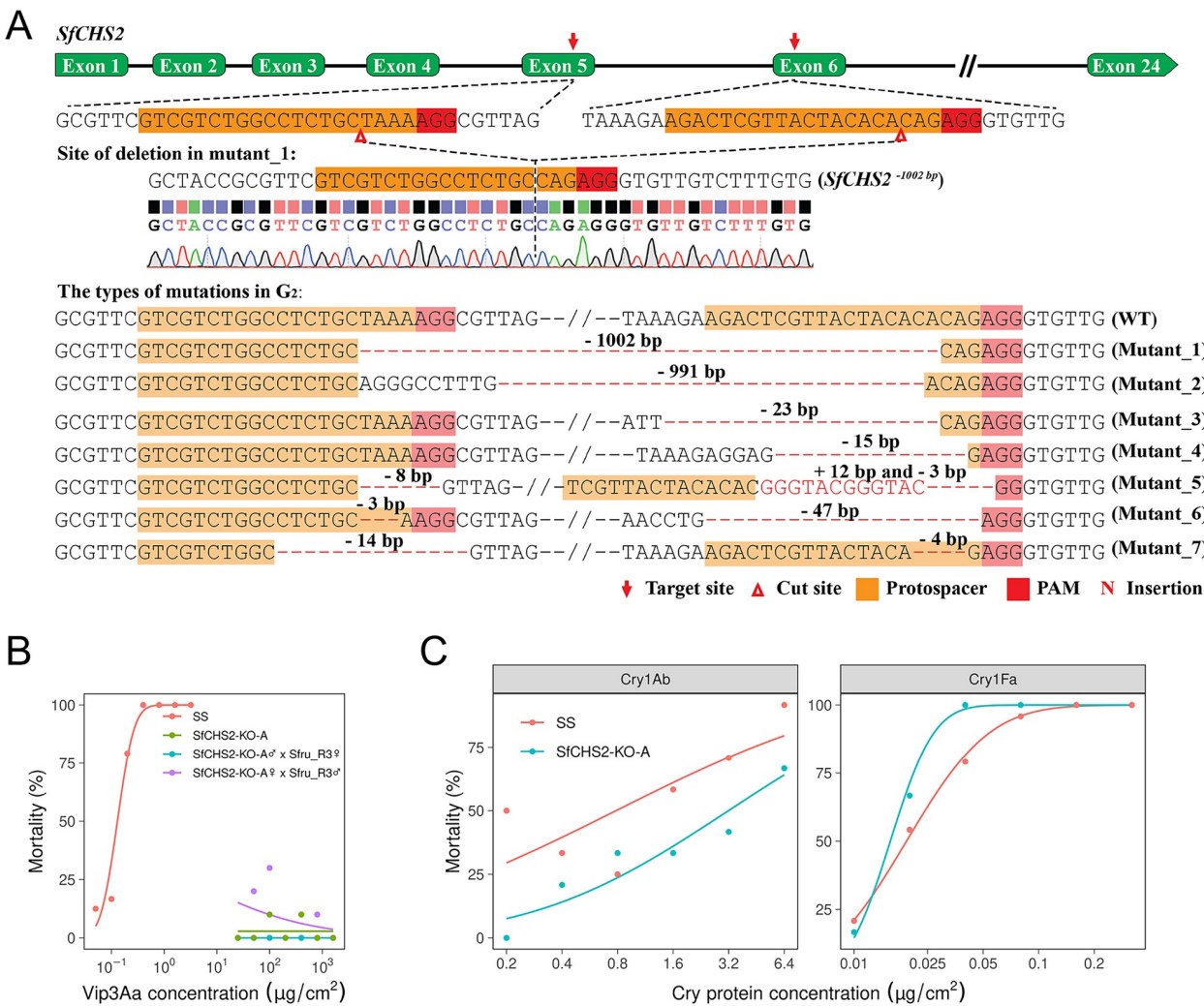

**Fig 3. Knockout of *SfCHS2* via CRISPR/Cas9 confers resistance to Vip3Aa in *S. frugiperda*.** (A) CRISPR/Cas9-mediated double sgRNA system and various types of mutations in G1 larvae identified through sequencing of individual PCR clones. Deleted bases are indicated as red dashes and inserted bases are indicated as red letters. The CRISPR target sites and the number of deleted and inserted bases (+, insertion;–, deletion) are shown. The chromatogram shows the sequence of the mutant isolated from a homozygous knockout larva in G2. (B) Log Vip3Aa concentration–response curves for SS, SfCHS2-KO-A, and the progeny of reciprocal crosses between SfCHS2-KO-A and Sfru_R3. (C) SfCHS2-KO-A did not show strong cross-resistance to Cry1Ab or Cry1Fa. The $EC_{50}$ did not differ significantly between strains for Cry1Ab (SfCHS2-KO-A: 3.2 [95% CI = 2.0–6.8] versus SS: 1.5 [1.0–2.1]; RR = 2.1) or Cry1Fa (SfCHS2-KO-A: 0.016 [0.013–0.019] versus SS: 0.019 [0.014–0.024], RR = 0.8), $n = 168$ larvae tested in each of the 4 bioassays. The data underlying this figure can be found in S3 Data and S9 Data. CI, confidence interval; RR, resistance ratio; SS, susceptible strain.

sequence is predicted to yield a truncated protein consisting of only the initial 247 amino acids of SfCHS2. The truncated sequences in SfCHS2-KO-B and SfCHS2-KO-C encode only the first 86 and 89 amino acids of CHS2, respectively (S6 Fig). Relative to SS, all 3 *S. frugiperda* knockout strains had a Vip3Aa RR >12,000 (S7 Table). In addition, the progeny of reciprocal crosses between Sfru_R3 and SfCHS2-KO-A were highly resistant to Vip3Aa (Fig 3B), which confirms that the recessive mutation conferring resistance to Vip3Aa affected the same locus (*SfCHS2*) in both strains. Like Sfru_R3, SfCHS2-KO-A did not show strong cross-resistance to Cry1Ab or Cry1Fa (Fig 3C). SlCHS2-KO of *S. litura* and MsCHS2-KO of *M. separata* had deletions of 207 bp between exons 3 and 4 and 366 bp between exons 4 and 5, respectively (S7 Fig).

Relative to their parent susceptible strains, the Vip3Aa RRs were >100,000 for SlCHS2-KO and >1,330 for MsCHS2-KO (S8 Table).

## Yaoer insertion in *SfCHS2* detected in a field-collected individual from China

Screening of the gDNA sequences of 540 *S. frugiperda* previously obtained from the field or laboratory in Africa, Asia, North America, and South America [43,44] detected the Yaoer insertion in 1 individual collected from the field in Jiangmen City, in the Guangdong province of southeastern China in 2020 (ID: GDJM14, S8 and S9 Figs and S9 Table). In this individual, the upstream and downstream insertion sites were covered by 4 and 6 reads, respectively, and no reads covered the junction between exon 21 and intron 21 (S9 Fig). By contrast, reads from the other 539 samples covered the exon 21-intron 21 junction site and neither insertion site (S9 Fig).

## Discussion

The results here from BSA, fine-scale mapping, and genetic linkage analyses show that 5,560-fold resistance to Vip3Aa in the laboratory-selected Sfru_R3 strain of *S. frugiperda* relative to its unselected parent strain SS is associated with a naturally occurring mutation disrupting the chitin synthase gene *SfCHS2*. This mutation entails insertion of a transposable element, the LTR retrotransposon Yaoer, and reduced abundance of wild-type transcripts of *SfCHS2*. Whereas previous findings show that transposable elements can cause resistance to Cry toxins [45–50], the results here are the first to identify this mechanism in resistance to Vip3Aa.

Knocking out the *CHS2* gene via CRISPR/Cas9 caused high levels of resistance to Vip3Aa in 3 strains of *S. frugiperda* and 1 strain each of 2 other major lepidopteran pests, *S. litura* and *M. separata*. These results imply that the CHS2 protein is essential for susceptibility to Vip3Aa in at least 3 lepidopteran species representing 2 tribes in the family Noctuidae. These findings also suggest that CHS2 might be important in susceptibility to Vip3Aa in other noctuids as well as more broadly in Lepidoptera.

The role of CHS2 proteins in mediating susceptibility to Vip3Aa remains to be determined. The finding that resistance to Vip3Aa in Sfru_R3 is associated with truncation of the C-terminal domain of SfCHS2 (C7) suggests that this domain plays a key role in susceptibility to Vip3Aa. Additional work is needed to test the hypothesis that mutations affecting CHS2 reduce binding of Vip3Aa in the larval midgut. This hypothesis is plausible because such reduced binding is the most common and most potent mechanism of insect resistance to Bt toxins [51]. For the Sfru_R3 strain analyzed here, the lack of strong cross-resistance to Cry1Ab, Cry1Ac, Cry1Fa, and Cry2Ab proteins associated with resistance to Vip3Aa is consistent with a target-specific mechanism rather a more general mechanism that alters processing or penetration of a wide spectrum of insecticidal Bt proteins. The recessive inheritance of resistance at a relatively high concentration of Vip3Aa (2 μg Vip3Aa per $cm^2$ diet) is also consistent with reduced binding of Vip3Aa as the mechanism of resistance [52]. Resistance to Vip3Aa was associated with reduced binding of Vip3Aa in the larval midgut for *H. zea* [53], but not in some other cases [10,54,55].

If future work supports the hypothesis that disruption of SfCHS2 is associated with reduced binding of Vip3Aa to midgut target sites, it will be useful to determine if CHS2 binds Vip3Aa or if it facilitates binding of Vip3Aa to one or more other receptors on the larval midgut membrane. Similar to some receptors for Cry proteins, CHS2 is a membrane-bound protein that contains multiple transmembrane domains and is highly expressed in the midgut. In addition,

like some Cry protein receptors, CHS2 occurs on the midgut brush border membranes and apical ends of microvilli in the lepidopteran *Manduca sexta* [39].

We found the Yaoer retrotransposon in the *SfCHS2* gene in the SfRu_R3 strain that was derived from the SS strain that originated in Ruili City in southwestern China and in a field-collected individual from Jiangmen City in southeastern China, 1,500 km from Ruili City. These results suggest that if *S. frugiperda* is exposed extensively to Vip3Aa in China, the Yaoer insertion might be important in field-evolved resistance there. However, we did not find Yaoer in *SfCHS2* in 539 of the 540 *S. frugiperda* we screened, with the total sample consisting of 158 individuals from the field in China and 382 individuals from the field and laboratory in other countries in Asia, Africa, and the Americas. Moreover, we did not detect Yaoer or reduced wild-type transcript abundance of *SfCHS2* in the laboratory-selected DH-R strain of *S. frugiperda*, which had 206-fold resistance to Vip3Aa and also was derived from the SS strain [31].

It is striking that the genetic basis of resistance differs between the 2 Vip3Aa-resistant strains derived by laboratory selection from the SS strain, which was started with approximately 100 *S. frugiperda* collected from field in 2019 [31]. This suggests that each of the 2 subsets of the SS strain selected in the laboratory with Vip3Aa to obtain a resistant strain contained one of the 2 different resistance alleles identified but not the other. Thus, field-evolved resistance to Vip3Aa in this pest, which would arise from selection of a much larger and more genetically diverse population than the SS strain, could entail a diverse genetic basis of resistance as seen with resistance to Cry1Ac in *Helicoverpa armigera* [56]. Accordingly, resistance to Vip3Aa in a single population might be conferred by mutations affecting one gene in some individuals and mutations affecting other genes in other individuals.

Our finding of one individual homozygous for the Yaoer insertion and no heterozygotes in the 158 field-collected insects screened from China yields an estimated frequency of 0.00633 for this mutation in China. Based on this estimate, the probability of at least 1 copy of the Yaoer insertion in the approximately 100 field-sampled individuals from China used to start the SS strain is approximately 0.72 ($1 - [1-0.00633]^{200}$). Thus, it is likely that at least 1 copy of the Yaoer insertion occurred in the field-collected individuals that started the SS strain, even though we cannot exclude the possibility that this mutation arose in the SS strain after it was established in the lab. The initial slow response of Sfru_R3 to selection with Vip3Aa followed by an exponential increase in the resistance ratio (S1 Table) is similar to the general pattern expected for the response to selection conferred by a recessive resistance allele that is rare initially [57].

Related work with 5 laboratory-selected strains of *H. zea* from the southern US highly resistant to Vip3Aa also shows variation among strains in the genetic basis of resistance [58]. Although the specific genes conferring resistance to Vip3Aa were not identified, interstrain complementation tests revealed 3 distinct genetic loci involved, with 1 locus implicated in 2 strains derived from the same field collection, a second locus implicated in 2 other strains from field collections in different states, and the third locus involved in just 1 strain [58].

One of the factors determining which resistance alleles are most likely to contribute to field-evolved resistance is whether they cause fitness costs in the absence of Bt proteins [59], which is especially important in governing the frequency of resistance alleles before pest populations are exposed extensively to a particular Bt protein. Disruption of the normal function of class B chitin synthases, such as SfCHS2, could cause fitness costs because these enzymes produce chitin for the peritrophic matrix, which is important for digestion and protection from pathogens [34,35,41,42]. Unexpectedly, survival on non-Bt maize was 1.4-fold higher for the Vip3Aa-resistant strain Sfru_R3 than its parent susceptible strain SS, implying a fitness benefit for this trait rather than a cost was associated with resistance to Vip3Aa. The presence of some wild-type SfCHS2 transcripts may reduce fitness costs in Sfru_R3. Also, the truncation of only the C-terminal domain encoded by mutant transcripts in Sfru_R3 might enable partial

functioning of SfCHS2. Nonetheless, we cannot exclude the possibility of fitness costs associated with resistance to Vip3Aa affecting development rate or other life history traits that were not evaluated here.

In Sfru_R3, total transcript abundance of SfCHS2 was not reduced relative to SS, which differs from some previous studies where RNAi was used to decrease total CHS2 transcript abundance [36,37,59]. RNAi suppression of *SfCHS2* (referred to as *SfCHSB*) by feeding bacteria producing dsRNA to second instar larvae reduced growth and survival to adulthood [36], whereas injecting third instar larvae with dsRNA produced some abnormal phenotypes but did not decrease survival after 24 or 48 h [37]. Although susceptibility to Cry1Ac did not differ between Sfru_R3 and SS in our study, Kim and colleagues [60] found that RNAi suppression of *SeCHS2* in *Spodoptera exigua* increased susceptibility to Cry1Ac and Cry1Ca. They did not evaluate the potential impact on susceptibility to Vip3Aa.

In conclusion, the results here demonstrate that CHS2 is essential for susceptibility to Vip3Aa in 3 major lepidopteran pests. These findings and the identification in a field-collected individual of the Yaoer retrotransposon insertion in *SfCHS2*, which is associated with resistance to Vip3Aa in the Sfru_R3 strain, have the potential to improve monitoring and management of resistance to Vip3Aa. It will be intriguing to determine if high levels of resistance to Vip3Aa in other pests and laboratory-selected strains of *S. frugiperda* not studied here are associated with insertion of Yaoer or other mutations disrupting CHS2. More extensive screening for mutations disrupting CHS2 in field-collected samples of *S. frugiperda* could also be useful, particularly in regions where transgenic crops producing Vip3a have been grown intensively for several years. Considered together with related work, the results here suggest that although the genetic basis of field-evolved resistance to Vip3Aa could be diverse, further investigation of mutations disrupting CHS2 could be particularly useful for sustaining the efficacy of Vip3Aa against pests.

## Materials and methods

### Insect strains

The susceptible strain SS (previously called DH-S [31]) of *S. frugiperda* was started from insects collected in 2019 from non-Bt maize fields in Ruili City, Dehong Prefecture, Yunnan Province, China (24°08′47″N, 97°82′33″E). Approximately 100 field-collected individuals were reared to pupae on non-Bt corn in the laboratory to start the strain [31]. SS was maintained in the laboratory without exposure to Bt toxins. The Vip3Aa-resistant strain, Sfru_R3, was developed from SS by selection on diet with Vip3Aa incorporated (see below). SS was the parent strain for the 3 mutant strains generated by CRISPR/Cas9 with the *CHS2* gene knocked out: SfCHS2-KO-A, SfCHS2-KO-B, and SfCHS2-KO-C. Susceptible strains of *S. litura* (Sl-SS) and *M. separata* (Ms-SS) were started with larvae collected in 2017 from non-Bt maize near Beijing in 2017. Sl-SS and Ms-SS were the parent strains for the mutant strains SlCHS2-KO and MsCHS2-KO, respectively.

### Toxin preparation and bioassay

The Vip3Aa protoxin used in all insect bioassays was purchased from Beijing Genralpest Biotech Research Co., Ltd. (www.genralpest.com), Beijing, China. For brevity, we refer to Vip3Aa protoxin as Vip3Aa. It was produced by the BL21 strain of *Escherichia coli*. Vip3Aa protein from cell lysates was purified on a HiTrap Chelating HP column (GE Healthcare, Freiburg, Germany) and was examined for purity by sodium dodecyl sulfate-polyacrylamide gel electrophoresis (SDS-PAGE) analysis. The Vip3Aa protein in the crude cell lysate was quantified by

densitometry of the stained band (approximately 88 kDa) on the SDS-PAGE gel. The crude cell lysate was stored at $-80°C$ until used for bioassays.

Diet overlay bioassays were conducted to assess responses to Vip3Aa. Diet was poured into 24-well plates and different concentrations of Vip3Aa applied to the surface and allowed to dry. First instar larvae were transferred to the wells. After 7 days, the number responding was scored as dead larvae plus live first and second instar larvae. We used probit analysis with SPSS 18.0 to calculate $EC_{50}$, which is the concentration (µg Vip3Aa per $cm^2$ diet) causing 50% of larvae to die or not reach third instar after 7 days and its 95% confidence interval (CI). $EC_{50}$ values with non-overlapping 95% CIs were considered significantly different. Resistance ratios were calculated by dividing the $EC_{50}$ of a strain or progeny from a cross by the $EC_{50}$ of SS, the Vip3Aa-susceptible strain.

## Selection of resistance to Vip3Aa in *S. frugiperda*

For the first selection to establish Sfru_R3, we exposed more than 10,000 first instar larvae from SS to diet containing 2.22 µg Vip3Aa per g, which was the $EC_{95}$ of SS. Five days later, about 400 larvae that developed to third instar were transferred to untreated diet and reared to adults. From the second generation onward, larvae remained on Vip3Aa-treated diet for the entire larval period. After each bout of selection, individuals that survived to adulthood on treated diet mated to produce the next generation. The concentration of Vip3Aa was gradually increased to 24 µg Vip3Aa per g diet in generation 16 (S1 Table).

## Bioassays on Bt and non-Bt maize plants

We conducted bioassays with maize (*Zea mays*) plants in a greenhouse at the Chinese Academy of Agricultural Sciences in Beijing. Seeds were obtained from the DBN Group (Beijing) for maize producing Vip3Aa (DBN9501), Cry1Ab (DBN9936), or Cry1Ab + Vip3Aa (DBN9936 × DBN9501), and non-Bt maize (Nonghua106). Maize was grown with 5 plants per pot in a greenhouse at about 26°C. Previous results showed that the Vip3Aa concentration in leaves (in µg Vip3Aa per g freeze-dried leaf powder) was 5.1 for DBN9501, 6.8 for DBN9936 × DBN9501, and 0.0 for DBN9936 and Nonghua [61]. We transferred first instar larvae to maize when the plants were about 30 cm high and had 4 leaves and 1 shoot. We measured survival until adult eclosion. For each type of maize and each insect strain (Sfru_R3 and SS), we conducted 3 replicates with 72 larvae per replicate ($n$ = 216 larvae for each combination of maize and insect strain). Measurements were taken from distinct samples; the same sample was not measured repeatedly.

## Inheritance of resistance to Vip3Aa

Larval response (%) in the diet bioassays was calculated as = 100% × (number of dead larvae + number of surviving first instar larvae) divided by the total number of larvae tested. Larval response at each Vip3Aa protein concentration was corrected based on the mortality observed on the untreated control diet using Abbott's method. Probit analysis was used to estimate the $EC_{50}$ and 95% CI as above. In addition, larval responses at 2 µg Vip3Aa per $cm^2$ diet were analyzed.

At each of 4 concentrations (0.25, 0.5, 1, and 2 µg Vip3Aa per $cm^2$ diet), we evaluated the dominance of resistance ($h$), which ranges from 0 (completely recessive resistance) to 1 (completely dominant resistance) [33]. We calculated $h$ as (Z—X) divided by (Y—X), where X, Y, and Z are the survival (1 –response) of larvae from SS, Sfru_R3, and $F_1$ progeny from the cross between SS and Sfru_R3. Because the $EC_{50}$ did not differ significantly between the

progeny from the 2 reciprocal $F_1$ crosses, we pooled the data from the 2 sets of progeny to calculate $h$.

To analyze inheritance of Vip3Aa resistance in Sfru_R3, we used bioassays with Vip3Aa to test larvae from the following crosses (a) $F_1$: $F_1a$ = Sfru_R3♂ × SS♀ and $F_1b$ = Sfru_R3♀ × SS♂; (b) $F_2$: $F_2a$ = $F_1a$ × $F_1a$ and $F_2b$ = $F_1b$ × $F_1b$; and (c) backcrosses: BCR1 = $F_1a$♂ × Sfru_R3♀, BCR2 = $F_1a$♀ × Sfru_R3♂, BCR3 = $F_1b$♂ × Sfru_R3♀, and BCR4 = $F_1b$♀ × Sfru_R3♂.

We tested the hypothesis that resistance is controlled by 1 locus with 2 alleles: S (susceptible) and R (resistant). If so, the backcross between RS and RR will produce progeny that are 50% RR and 50% RS, the backcross between RS and SS will produce progeny that are 50% SS and 50% RS, and the $F_2$ are expected to consist of 25% RR, 50% RS, and 25% SS. With completely recessive resistance (which occurred at 2 μg Vip3Aa per $cm^2$ diet, S4 Table), only RR are expected to survive, yielding expected survival of 50% for the backcross larvae and 25% for the $F_2$.

## Whole-genome sequencing (WGS)-based bulked segregant analysis (BSA)

For resistance mapping, a single pair cross (male Sfru_R3 × female SS) was conducted to generate $F_1$ progeny. The head and thorax of the male and female ($F_0$) were snap-frozen in liquid nitrogen and stored at −80°C. The $F_1$ progeny were raised on a toxin-free diet and mated to produce $F_2$ progeny; 480 $F_2$ neonates were placed on a high Vip3Aa diet (4.0 μg/$cm^2$) and 480 $F_2$ neonates were placed on a low Vip3Aa diet (0.1 μg/$cm^2$) for 5 days. $F_2$ progeny on the high Vip3Aa diet that molted to the third instar after 5 days ($n$ = 89) were classified as resistant ($F_2$-R). $F_2$ progeny on the low Vip3Aa diet that had not yet molted to the third instar after 5 days ($n$ = 98) were classified as susceptible ($F_2$-S). These 2 groups were reared to the fifth instar on untreated diet. Each larva was snap-frozen and stored at −80°C. DNA was isolated separately from Sfru_R3, SS, $F_2$-R ($n$ = 50) and $F_2$-S ($n$ = 50) samples. Equal amounts of DNA from each individual $F_2$-S larva were mixed, resulting in a bulked (pooled) DNA sample representing the $F_2$-S group; the $F_2$-R group was pooled in a similar manner. The DNA samples were sent to Biomarker (Beijing, China) for whole-genome sequencing on the Illumina NovaSeq 6000 platform. The requested sequence coverage for parents and $F_2$ progeny were 20× and 100×, respectively.

The raw reads were filtered by using trimmomatic [62], then clean reads were mapped onto the most recent *S. frugiperda* reference genome at GenBank (AGI-APGP_CSIRO_Sfru_2.0, GCF_023101765.2) using Burrows–Wheeler aligner (BWA) [63]. Then, GATK (v 4.4.0.0) [64] was used to perform variant calling, including single nucleotide polymorphisms (SNPs) and indels. We filtered out SNP sites that showed (a) read support less than 4; (b) no genotype difference between the $F_0$ susceptible (SS) and $F_0$ resistant male (Sfru_R3); or (c) genotypes of $F_2$ that did not originate from the $F_0$. For each remaining SNP locus, we calculated ΔSNP as the frequency of alleles originating from the Sfru_R3 male $F_0$ minus the frequency of alleles from SS. Thus, ΔSNP ranged from 0 to 1, with 1 indicating the maximum association with resistance to Vip3Aa. We calculated the ΔSNP index as the mean ΔSNP for each 1 kb window. Code for the BSA analysis is at https://doi.org/10.5281/zenodo.11395059 or https://github.com/lumingzou/Yaoer-BSA.

## Fine-scale mapping

For fine-scale mapping of Vip3Aa resistance, 50 $F_1$ progeny of a SS female × Sfru_R3 male cross were intercrossed to form the $F_2$; 50 $F_2$ progeny were intercrossed to form the $F_3$, and so on for 7 generations. Neonates from the $F_7$ generation were divided into 2 groups: >10,000 larvae on high Vip3Aa diet (24.0 μg/$cm^2$) and 2,400 larvae on low Vip3Aa diet (0.1 μg/$cm^2$) for

5 days. The 214 largest survivors on the high Vip3Aa diet formed the $F_7$-R group and the 109 slowest-growing survivors on the low Vip3Aa diet formed the $F_7$-S group. Both groups were reared to the fifth instar on toxin-free diet. The genomic DNA of 157 third instar $F_7$-R larvae and the original parents was isolated individually using the DNeasy Blood & Tissue Kit (Qiagen, Germany). The midguts of $F_7$-R ($n = 16$) and $F_7$-S ($n = 16$) were homogenized and stored in TRIzol reagent (Invitrogen, Carlsbad, United States of America) at $-20°C$ for RNA isolation.

Based on the genome sequence and homozygous SNPs differentiating SS and Sfru_R3, we designed genetic markers in the exons of 8 evenly spaced genes in the candidate genomic region. Specific primers (S10 Table) for the 8 SNP markers were designed and used to investigate the genotypes of 157 $F_7$-R individuals by PCR amplification and sequencing. Denoting SNP alleles from Sfru_R3 as r and those from SS as s, the numbers of rr or ss homozygotes and rs heterozygotes for each locus marker were recorded.

## RNA sequencing (RNA-Seq)

The midguts of fifth instar larvae were collected, rapidly frozen in liquid nitrogen, and stored at $-80°C$. Total RNA was extracted using the TRIzol reagent according to the manufacturer's instructions. The RNA was sent to BerryGenomics (Beijing, China) for library construction and sequencing. Raw sequence reads were assessed for quality using FastQC. Trimming of adapter sequences and low-quality bases was performed using trimmomatic. The processed reads were aligned to the *S. frugiperda* reference genome (GenBank Accession GCF_023101765.2) using HISAT2. StringTie was employed to assemble transcripts and quantify gene expression levels. StringTie-generated GTF files containing transcript information were processed to create gene-level FPKM tables.

## Genomic DNA isolation, RNA extraction, cDNA synthesis, and RT-qPCR analysis

The gDNA of single fourth instar larvae was isolated for detection using the AxyPrep Multisource Genomic DNA Miniprep Kit (Axygen, New York, USA) according to the manufacturer's specifications. PCR reactions were performed in a S1000 Thermal Cycler (Bio-Rad) using the 2×Taq Plus Master Mix II (Vazyme, Nanjing, China) as follows: initial denaturation 95°C for 3 min, followed by 35 cycles of 95°C for 15 s, 55°C for 15 s, and 72°C for 30 s, and a final extension of 5 min at 72°C.

Total RNA was extracted from *S. frugiperda* midgut tissues using TRIzol reagent (Invitrogen) following the manufacturer's instructions, and 1 μg total RNA was used to synthesize the first-strand cDNA using the Revert Aid First Strand cDNA Synthesis Kit (Thermo Fisher Scientific, Waltham, Massachusetts, USA). The relative expression levels of targeted genes were quantified by RT-qPCR with AceQ Universal SYBR qPCR Master Mix (Vazyme, Nanjing, China) in a Bio-Rad CFX Connect Real-Time System (Bio-Rad, USA). The reaction was performed in a final volume of 10 μl containing 5 μl of 2×AceQ Universal SYBR qPCR Master Mix, 1 μl of cDNA and 0.5 μl of each primer (10 μm). The reaction conditions were as follows: initial denaturation at 95°C for 2 min, followed by 40 cycles at 95°C for 5 s and 60°C for 30 s. A melting curve analysis was performed after the amplifications to determine the Tm of the amplicons as a quality check. The glyceraldehyde-3-phosphate dehydrogenase gene (GAPDH) was used as an internal control. Primers were GAPDH_F: 5′-CGG TGT CTT CAC AAC CA C AG-3′ and GAPDH_R: 5′-TTG ACA CCA ACG ACG AAC AT-3′. For quantitative analysis of total transcripts of *SfCHS2*, the primers were SfCHS2-q-F1: 5′-TGT TCG TGC TCG TCA TCT TC-3′ and SfCHS2-q-R1: 5′-ACC GAT AGG TTC AGC GT TA-3′. For

quantitative analysis of wild-type transcripts of *SfCHS2*, the primers were SfCHS2-q-F2: 5′– GCC ATG TTG TTC CAT CGC CT–3′ and SfCHS2-q-R2: 5′–AGT CGT CGG TGT TCA GAC GT–3′. Quantitative analysis of gene expression was calculated using the $2^{-\Delta Ct}$ or $2^{-\Delta\Delta Ct}$ method. Primer locations are depicted in Fig 2D. PCR conditions were initial denaturation 95˚C for 3 min, followed by 35 cycles of 95˚C for 15 s, 55˚C for 15 s, and 72˚C for 30 s.

## Sanger sequencing of the *SfCHS2* genomic sequence in SS and Sfru_R3

For genomic sequencing of *SfCHS2* in SS and Sfru_R3, a total of 17 pairs of primers (S10 Table) across the 5′ UTR and CDS region of *SfCHS2* were designed on the basis of the genomic sequence of *SfCHS2* from the reference genome assembly (AGI-APGP_CSIRO_Sfru_2.0, GCF_023101765.2). The PCR products were directly Sanger sequenced (Sangon Biotech Co., Ltd., Shanghai, China). To assure the accuracy of the full-length genomic sequence of SfCHS2, primers for the overlapping region with a range of 100 to 500 bp between each 2 adjacent PCR products were designed. All the primers for amplification of *SfCHS2* genomic sequence were designed based on the exon sequence. The length of all the PCR products ranged from 770 to 1,550 bp. The gDNA from SS and Sfru_R3 were isolated as described above. PCR reactions were performed in a S1000 Thermal Cycler (Bio-Rad) using the 2×Taq Plus Master Mix II (Vazyme, Nanjing, China) as follows: initial denaturation 95˚C for 3 min, followed by 35 cycles of 95˚C for 15 s, 55˚C for 15 s, and 72˚C for 30 to 60 s, and a final extension of 5 min at 72˚C. For the amplification of the genomic sequence across intron 21 of SfCHS2 in Sfru_R3, the extension time was lengthened to 3, 5, and 10 min, since no PCR products were obtained for Sfru_R3 under a shorter extension time. The PCR products (approximately 7 kb) between exon 21 and exon 22 from Sfru_R3 were also directly Sanger sequenced as described above.

## Detection of *SfCHS2* transcripts via PCR and isoform-sequencing (Iso-Seq)

For PCR detection, specific primers (S10 Table) were designed to detect the transcripts in SS and Sfru_R3. The total RNA of midgut tissue from individual larvae were isolated and used to synthesize cDNA as described above. PCR reactions were performed as described above.

For Iso-Seq, the total RNA of midguts from fifth instar larvae of SS and Sfru_R3 were isolated and sent to BerryGenomics (Beijing, China) for library construction and sequencing. For each sample, 200 ng of total RNA was input for cDNA synthesis and amplification using NEB Next Single Cell/Low Input cDNA Synthesis & Amplification Module (New England BioLabs Inc., Ipswich, Massachusetts, USA) following the manufacturer's standard protocol. A total of 13 PCR cycles were used to generate sufficient quantities of cDNA for PacBio Iso-Seq library preparations. SMRTbell Iso-Seq libraries was constructed using Express Template Prep 2.0 (Pacific Biosciences, Menlo Park, California, USA) by following the manufacturer's Iso-Seq Express Template Preparation protocol. The SS and Sfru-R3 Iso-Seq libraries were run on a single sequel system SMRT Cell using sequencing chemistry 3.0 with 4-h pre-extension and 20-h movie time. Raw reads were processed into circular consensus sequence (CCS) reads as per the manufacturer's standard pipeline (SMRT Link version 8.1). The Pacific Biosciences toolkit (https://github.com/PacificBiosciences/pbbioconda) was utilized for Iso-Seq data analysis. Initially, raw reads underwent primer removal using lima (v2.7.1) and subsequent refinement involving polyA removal using isoseq3. Following this preprocessing, the clean reads underwent clustering via isoseq3 cluster to generate transcriptional sequences. The isoforms of *SfCHS2* were then extracted using NCBI-BLAST (v2.11.0) and aligned to the reference sequence using Minimap2 (v2.26) [65].

## Genetic linkage analysis

To test the association between retrotransposon insertion in *SfCHS2* and Vip3Aa resistance, we firstly investigated the genotype of *SfCHS2* in $F_7$-R ($n = 16$) and $F_7$-S group ($n = 16$) using a pair of specific primers (e21-F: 5′-GCC ATG TTG TTC CAT CGC CT-3′/e22-R: 5′-AGT CGT CGG TGT TCA GAC GT-3′) which across the 21st intron and was predicted to amplify a 467 bp fragment of wild-type *SfCHS2* under the following condition: initial denaturation 95°C for 3 min, followed by 35 cycles of 95°C for 15 s, 55°C for 15 s, and 72°C for 30 s. Parallelly, a pair of specific primers (e21-F: 5′-ACC CAA GAC TAC TTA ACG CT-3′/e21-R: 5′-TTT GGT GGT GGA CAG CAG AT-3′) in exon 21 were used as the positive control. To further test the genetic association between resistance to Vip3Aa and reduced wild-type transcripts of *SfCHS2*, a pair of specific primers (SfCHS2-q-F2/R2) flanking the intron 21 were designed to evaluate the wild-type transcripts of *SfCHS2* in SS, Sfru_R3, $F_7$-S, and $F_7$-R. A pair of primers (SfCHS2-q-F1/R1) at exons 20 and 21 was designed to evaluate the relative abundance of total transcripts of *SfCHS2* in SS, Sfru_R3, $F_7$-S, and $F_7$-R. The reaction conditions for RT-qPCR were as mentioned above. The abundance of *SfCHS2* transcripts in Sfru_R3 and individuals from $F_7$-S and $F_7$-R were normalized to the fold value of $2^{-\Delta Ct}$ relative to SS.

## CRISPR/Cas9 knockouts

The CRISPR/Cas9 system was used to create deletions in the *SfCHS2* gene from the SS strain. Briefly, 3 single-guide RNAs (sgRNA3, sgRNA5, and sgRNA6) were designed using the sgRNAcas9 (V3.0) software (www.biootools.com/software). The template DNA was made with PCR-based fusion of 2 oligonucleotides with the T7 promoter (Target F: 5′-TAA TAC GAC TCA CTA TAG + the target sequence; Target R: 5′-TTC TAG CTC TAA AAC + the target sequence reverse complement). The target sequences plus PAMs were as follows: sgRNA3 targeting exon 3: sf-chs-sgR3+PAM = 5′-GGA TCT GCG GTT GTG TCT AA GGG-3′; sgRNA5 targeting exon 5: sf-chs-sgR5+PAM = 5′-GTC GTC TGG CCT CTG CTA AA AGG-3′, and sgRNA6 targeting exon 6: sf-chs-sgR6+PAM = 5′-AGA CTC GTT ACT ACA CAC AG AGG-3′. For *S. litura*, the target sequences plus PAMs were sgRNA3 targeting exon 3: Slit_v3-sgR3+PAM = 5′-GGA TCG GCG GTT GTG TCT AA GGG-3′ and sgRNA4 targeting exon 4: Slit_v3-sgR+PAM = 5′-AGA GCG TGT GAC ATG GCT GT GGG-3′. For *M. separata*, the target sequences plus PAMs were sgRNA4 targeting exon 4: Msep_v3-sgR4+PAM = 5′-GCG TAT TGG TTT CTC TCG GC GGG-3′ and sgRNA5 targeting exon 5: Msep_v3-sgR5+PAM = 5′-CCA TTG CAA AAT CTC CGC GA GGG-3′. In vitro transcription was performed with the GeneArt Precision gRNA Synthesis Kit (Thermo Fisher Scientific, Waltham, Massachusetts, USA) according to the manufacturer's instructions. The Cas9 protein (GeneArt Platinum Cas9 Nuclease) was purchased from Thermo Fisher Scientific.

For embryo collection and microinjection, freshly laid eggs (within 2 h after oviposition) were immersed in 1% sodium hypochlorite solution for 10 s and washed off from the oviposition gauze, and finally rinsed with distilled water. The eggs were placed on a microscope slide and fixed with double-sided adhesive tape. About 2 nL of a mixture of sgRNAs (250 ng/μl) and Cas9 protein (150 ng/μl) was injected into individual eggs using the Nanoject III (Drummond Scientific, Broomall, Pennsylvania, USA). The microinjection was completed within 2 h.

Genomic DNA was isolated from a hind leg of $G_0$ adults prior to mating. Primer pairs used to detect specific deletions were as follows: sf-chs-F1 5′-AGC TCA AGA GGC AAA AGG AT-3′ in exon 3 and sf-chs-R1 5′-AGC TAA TTG AGT GGC TCC CT-3′ in intron 5 for SfCHS2-KO-B; sf-chs-F2 5′-GCC TTC GTA GTA GAC ACC CT-3′ in exon 5 and sf-chs-R2 5′-CAT GAA CTT TGT AGA AGC GCT C-3′ in exon 6 for SfCHS2-KO-A; and sf-chs-

F1 5′-AGC TCA AGA GGC AAA AGG AT-3′ in exon 3 and sf-chs-R2 5′-CAT GAA CTT TGT AGA AGC GCT C-3′ in exon 6 for SfCHS2-KO-C. For *S. litura*, the primer pairs used to detect deletions for SlCHS2-KO were Slit_v3-F1 5′-AGG ATG GAA TCT GTT CCG AG-3′ in exon 3 and Slit_v3-R1 5′-TTG CAA AAA CGT AGG CTT CG-3′ in exon 4. Subsequently, PCR products of the region surrounding the target sites were sequenced to determine the exact mutation types in the $G_0$. Selected $G_0$ mutants were crossed to SS to generate $G_1$ progeny. Ultimately, 13 homozygotes with SfCHS2 deletions were mated to establish a homozygous *SfCHS2* knockout strain designated SfCHS2-KO-A.

## Detection of resistance allele of *SfCHS2* with Yaoer insertion in the laboratory and field population of *S. frugiperda*

To identify the Yaoer insertion in the laboratory and field population of *S. frugiperda*, the WGS reads from 540 samples collected globally (S9 Table) were aligned with insertion site sequences. These sequences consist of 600 bp, representing 300 bp extensions in each direction from the insertion sites. The alignment was performed using BWA. Insertion sites covered by more than 3 reads were classified as positive insertions.

## Supporting information

**S1 Table. Selection of the Sfru_R3 strain for resistance to Vip3Aa.**
(DOCX)

**S2 Table. Survival from first instar to adult eclosion of *S. frugiperda* strains Sfru_R3 and SS on Bt and non-Bt maize.**
(DOCX)

**S3 Table. Responses of strains SS and Sfru_R3 to Cry1Ab, Cry1Ac, Cry1Fa, and Cry2Ab.**
(DOCX)

**S4 Table. Dominance of Vip3Aa resistance in *Spodoptera frugiperda* based on survival of Sfru_R3, SS, and their $F_1$ progeny.**
(DOCX)

**S5 Table. Observed mortality at 2 μg Vip3Aa per $cm^2$ diet versus mortality expected assuming resistance is controlled by a recessive allele at a single locus.**
(DOCX)

**S6 Table. Normalized midgut transcript abundance for genes from 8.17 to 8.64 Mb on chromosome 1.**
(DOCX)

**S7 Table. Responses to Vip3Aa of three SfCHS2-knockout strains and progeny from reciprocal crosses between a knockout strain SfCHS2-KO-A and Sfru_R3.**
(DOCX)

**S8 Table. Responses to Vip3Aa of *CHS2*-knockout strains of *S. litura* and *M. separata*.**
(DOCX)

**S9 Table. Individuals of *S. frugiperda* screened for the Yaoer insertion in *SfCHS2*.**
(DOCX)

**S10 Table. Primers for fine-mapping, amplification of *SfCHS2* genomic sequence, and *SfCHS2* transcripts detection.**
(DOCX)

**S1 Fig. Sample preparation strategy for bulked segregant analysis (BSA).** For resistance mapping, a single-pair cross was conducted between a male moth from the Sfru_R3 colony and a female moth from the SS colony to generate $F_1$ progeny. The $F_1$ progeny were raised on a normal diet, resulting in the production of $F_2$ progeny. A total of 960 neonate larvae (480 for both selections) from the $F_2$ generation were subjected to 2 different diets: a high Vip3Aa diet (4.0 μg/cm$^2$) and a low Vip3Aa diet (0.1 μg/cm$^2$) for a duration of 5 days. In the case of high Vip3Aa concentration, the individuals that developed into the third instar after 5 days of exposure were classified as resistant to Vip3Aa ($F_2$-R) ($n = 89$). Conversely, for the low Vip3Aa concentration, the individuals that were still <third instar were considered susceptible to Vip3Aa ($F_2$-S) ($n = 75$). Following this classification, both the resistant and susceptible larvae from $F_2$ generation were transferred to a normal diet until they reached the fifth instar stage. (DOCX)

**S2 Fig. Insect chitin synthases.** Neighbor-joining tree of Class A and Class B chitin synthases of insects. Class A enzymes make chitin for the exosekeleton; Class B enzymes make chitin for the peritrophic matrix. Both catalyze reversible elongation of the chitin chain (n to n+1) by addition of GlcNAc donated by UDP-GlcNAc: UDP-N-acetyl-D-glucosamine + [1,4-(N-acetyl-beta-D-glucosaminyl)]n ⇌ UDP + [1,4-(N-acetyl-beta-D-glucosaminyl)]n+1. *Aedes aegypti*: AaCHS1 = XP_021704891.1, AaCHS2 = XP_001651163.1. *Drosophila melanogaster*: DmCHS1 = AAG22215.3 (krotzkopf verkehrt), DmCHS2 = AAF51798.2. *Manduca sexta*: MsCHS1 = AAL38051.2, MsCHS2 = AAX20091.1. *Tribolium castaneum*: TcCHS1 = NP_001034491.1, TcCHS2 = NP_001034492.1. *Spodoptera frugiperda*: SfCHS1 = XP_050552783.1, SfCHS2 XP_050552796.1. *Spodoptera litura*: SlCHS1 = XP_022820392.1, SlCHS2 XP_050552796.1. The data underlying this figure can be found in S4 Data. (DOCX)

**S3 Fig. Insertion site of LTR retrotransposon Yaoer and alternative splicing of SfCHS2.** (A) By comparison to the wild-type sequence, it is possible to deduce the target site duplication (TSD) typically created by transposable element insertion, which is GAAGG in this case. This represents the last 4 bases of exon 21 and the first base of intron 21. (B) In the inserted allele, the LTRa immediately follows the first GAAGG, and since the first base of the LTRa is a T, a GT corresponding to the 5′-GU donor site of the pre-mRNA is created immediately after exon 21. The second GAAGG occurs immediately after the second LTR. (C) In accordance with the splicing rules, the entire insert could be spliced out, restoring the wild-type coding sequence (starting with the first G remaining from intron 21, through LTRa, Yaoer, LTRb and to the end of wild-type intron 22 with its 3′-AG acceptor site). (D) However, the inserted allele has the target site duplication GAAGG immediately after the second LTR, immediately preceding the T which is the second base of the wild-type intron 21. This restores the original 5′ donor splice site of intron 21. Therefore, if just the original intron 21 were spliced out, removing its 3′-AG acceptor site, this could block further splicing of the LTR retrotransposon from the pre-mRNA. In that case, read-through from exon 21 in the mature mRNA would encounter an in-frame stop codon in LTRa, leading to translation of a truncated protein of 1,355 residues, where the last 12 residues (VSILFTLLIYLL*) are encoded by the LTR. (E) Alternatively, splicing out just the entire Yaoer element, if possible, would result in another truncated protein ending in VS* due to read-through from exon 21 to intron 21. (DOCX)

**S4 Fig. Predicted structures of wild-type and mutant SfCHS2 proteins.** (A) Wild-type protein. (B) Mutant protein in Sfru_R3 alternatively spliced because of the Yaoer retrotransposon insertion. The C-terminal lumenal domain (C7) is truncated. (C) Knockouts generated by CRISPR/Cas9. The domain structure and numbers of residues in transmembrane domains and intervening loops were predicted using Phobius (https://phobius.sbc.su.se/). Transmembrane domains are numbered in red.
(DOCX)

**S5 Fig. Transcripts of *SfCHS2* in SS and Sfru_R3.** (A) *SfCHS2* transcript detection via PCR and Sanger sequencing. Both wild-type and mutant *SfCHS2* transcripts with Yaoer were detected in Sfru_R3. Furthermore, the transcripts including both Yaoer and intron 21 were detected in Sfru_R3 by amplification products generated by primer pairs of 11F7/ Nei R5 and Nei F2/11 R4. (B) *SfCHS2* transcripts detection via Iso-Seq. In SS, multiple *SfCHS2* transcripts without Yaoer were detected. In contrast, both *SfCHS2* transcripts with and without Yaoer were detected in Sfru_R3. In addition, the transcripts of *SfCHS2* that would have spliced out the Yaoer element while including the intron21 were undetected in Sfru_R3. The data underlying this figure can be found in S1 Raw Images.
(DOCX)

**S6 Fig. Knockouts of *SfCHS2*. (A) SfCHS2-KO-B. (B) SfCHS2-KO-C.** CRISPR/Cas9-mediated double sgRNA system and various types of mutations in $G_1$ larvae identified through sequencing of individual PCR clones. Deleted bases are indicated as red dashes, and inserted bases are indicated as red letters. The CRISPR target sites and the number of deleted and inserted bases (+, insertion;–, deletion) are shown. The chromatogram shows the sequence of the mutant isolated from a homozygous knockout larva in $G_2$.
(DOCX)

**S7 Fig. Knockouts of *CHS2*. (A) *Spodoptera litura*. (B) *Mythimna separata*.** CRISPR/Cas9-mediated double sgRNA system and various types of mutations in $G_1$ larvae identified through sequencing of individual PCR clones. Deleted bases are indicated as red dashes, and inserted bases are indicated as red letters. The CRISPR target sites and the number of deleted and inserted bases (+, insertion;–, deletion) are shown. The chromatogram shows the sequence of the mutant isolated from a homozygous knockout larva in $G_2$.
(DOCX)

**S8 Fig. Locations of field and laboratory populations of *S. frugiperda* screened for the Yaoer insertion in *SfCHS2*.** A total of 540 *S. frugiperda* individuals collected from the field or laboratory were re-sequenced and mapped with the junctions of exon 21 or intron 21 and the Yaoer insertion (S9 Fig). One individual collected from the field of China in 2020 showed positive mapping. The red star shows the site for the positive mapping individual. The black circles show all other collection sites. See S9 Table for details of each sample.
(DOCX)

**S9 Fig. Detection of the Yaoer retrotransposon in *SfCHS2* in an individual collected from the field in China during 2020.** (A) Read coverage in the sample with the Yaoer insertion. (B) Read coverage in a wild-type sample. The insertion site sequences are depicted as colored bars. The reads from R2 are shown as gray horizontal bars and from R1 as horizontal white bars. Colored vertical bars within reads indicate variants.
(DOCX)

**S1 Data. Genome-wide mean ΔSNP index for each 1 kb window.**
(TXT)

**S2 Data. Total and wild-type transcripts of *SfCHS2* in SS, Sfru_R3, $F_7$-S, and $F_7$-R individual larvae.**
(XLSX)

**S3 Data. Response of SfCHS2-KO and $F_1$ progeny of cross with Sfru_R3 to Vip3Aa toxin and cross-resistance to other Bt toxins.**
(XLSX)

**S4 Data. Alignment file for CHS2 tree.**
(TXT)

**S5 Data. Bioassays of *S. frugiperda* during Vip3Aa-resistance selection process.**
(XLSX)

**S6 Data. Bt corn bioassays of Sfru_R3.**
(XLSX)

**S7 Data. Bioassays of Sfru_R3 to various Cry toxins.**
(XLSX)

**S8 Data. Bioassays of $F_1$, $F_2$, BC progeny of Sfru_R3 and SS at certain concentration of Vip3Aa.**
(XLSX)

**S9 Data. Bioassays of SfCHS2-KO and $F_1$ progeny of cross with Sfru_R3.**
(XLSX)

**S10 Data. Bioassays of CHS2-knockout strains of *S. litura* and *M. separata*.**
(XLSX)

**S1 Raw Images. Original, uncropped, and minimally adjusted images of all blots and gels reported in the article.**
(PDF)

## Author Contributions

**Data curation:** Zhenxing Liu, Chongyu Liao, Luming Zou, Lei Zhang, David G. Heckel.

**Formal analysis:** Zhenxing Liu, Chongyu Liao, Luming Zou, Minghui Jin, Bruce E. Tabashnik, David G. Heckel.

**Funding acquisition:** Minghui Jin, Kongming Wu, Yutao Xiao.

**Investigation:** Zhenxing Liu, Chongyu Liao, Yinxue Shan, Hui Yao, Zhuangzhuang Liu, Na Wang, Anjing Li, Kaiyu Liu.

**Project administration:** Kongming Wu, Yutao Xiao.

**Resources:** Yinxue Shan, Hui Yao, Lei Zhang.

**Supervision:** Kongming Wu, Yutao Xiao.

**Visualization:** Zhenxing Liu, Chongyu Liao, Luming Zou, David G. Heckel.

**Writing – original draft:** Chongyu Liao.

**Writing – review & editing:** Yudong Quan, Peng Wang, Bruce E. Tabashnik, David G. Heckel, Yutao Xiao.

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
