## [Editor Report · Decision Letter 0]

7 Mar 2024

Dear Dr Xiao, 

Thank you for submitting your manuscript entitled "Retrotransposon-mediated variation of a chitin synthase gene confers insect resistance to Bacillus thuringiensis Vip3Aa toxin" for consideration as a Research Article by PLOS Biology. I would like to apologize for the delay in giving you an initial response. 

Your manuscript has now been evaluated by the PLOS Biology editorial staff, as well as by an academic editor with relevant expertise, and I am writing to let you know that we would like to send your submission out for external peer review.

Once your full submission is complete, your paper will undergo a series of checks in preparation for peer review. After your manuscript has passed the checks it will be sent out for review. To provide the metadata for your submission, please Login to Editorial Manager (https://www.editorialmanager.com/pbiology) within two working days, i.e. by Mar 09 2024 11:59PM.

Kind regards,

Melissa

Melissa Vazquez Hernandez, Ph.D.

Associate Editor

PLOS Biology

---

## [Decision Letter · Decision Letter 1]

23 Apr 2024

Dear Dr Xiao,

Thank you for your patience while your manuscript "Retrotransposon-mediated disruption of a chitin synthase gene confers insect resistance to Bacillus thuringiensis Vip3Aa toxin" went through peer-review at PLOS Biology. Your manuscript has now been evaluated by the PLOS Biology editors, an Academic Editor with relevant expertise, and by two independent reviewers.

As you will see in the reports, all reviewers are positive about the study, but still have some concerns. Specifically, Reviewer #1 requires clarifications regarding the resistance evolution during the selection process. Reviewer #2 has some questions about the origin of the mutation obtained in the lab, the fitness costs of the CRISPR KO, and the methods describing the transposon. Addressing all the concerns is essential for further consideration of your manuscript for publication in PLOS Biology.

IMPORTANT: We think that it would be better to consider your paper as a Short Report. As your manuscript is already concise, no re-formatting is needed at this stage. Please select "Short Reports" as the article type when you submit your revised version. If our system does not allow you to change the article type, we can do this on your behalf after re-submission.

**IMPORTANT - SUBMITTING YOUR REVISION**

*Resubmission Checklist*

*Published Peer Review*

*PLOS Data Policy*

Please note that as a condition of publication PLOS' data policy (http://journals.plos.org/plosbiology/s/data-availability) requires that you make available all data used to draw the conclusions arrived at in your manuscript. If you have not already done so, you must include any data used in your manuscript either in appropriate repositories, within the body of the manuscript, or as supporting information (N.B. this includes any numerical values that were used to generate graphs, histograms etc.). For an example see here: http://www.plosbiology.org/article/info:doi%2F10.1371%2Fjournal.pbio.1001908#s5

*Blot and Gel Data Policy*

Sincerely,

Melissa

Melissa Vazquez Hernandez, Ph.D.

Associate Editor

PLOS Biology

REVIEWERS' COMMENTS

Reviewer #1: 

This manuscript reports the finding that chitin synthase 2 (CHS2) mutation confers high level resistance to Bt toxin Vip3Aa in insects. The authors obtained a strain of Spodoptera frugiperda, Sfru_R3, highly resistant to Vip3Aa (resistance ratio=5560) by selecting a susceptible colony with Vip3Aa for 17 generations. Using BSA (bulk segregant analysis) and Amp-seq based mapping, the resistance was mapped to the SfCHS2 locus. By DNA sequencing of the genes and RNA-seq to identify the midgut-expressed genes in locus, the authors identified the resistance associated mutation in the SfCHS2. The mutation was an insertion of a transposon in the SfCHS2 gene. The high level resistance to Vip3Aa was reproducible by introducing knockout mutations in the CHS2 gene in S. frugiperda and also in two other insects, S. litura and M. separata. The finding of an important role of CHS2 in toxicity of a Bt toxin in insects is new to the field of Bt research. The techniques and experimental designs in this study to identify the resistance conferring gene mutation are sound.

Other comments:

1. It is interesting to note that in a recent report from this same group a moderate level Vip3Aa resistant strain was selected from the very same susceptible colony (DH-S) by selection with Vip3Aa and the genetic mechanism for resistance is independent of CHS2 mutation. As Bt resistance development in insect populations in agriculture is a process of selection of resistant alleles. Even though the selection in this study is in lab, it would be interesting to readers, if authors could provide any insights to the observations that two similar selections from the same susceptible colony resulted in two very different resistant strains with different genetic mechanisms and the two mechanisms were not co-selected in a strain.

2. Another interesting observation from the selection process is that the resistance trait selected in this study was very high level resistance controlled by a single gene mutation, but the resistance level appeared to increase rather slowly in the selection process. As the dominance H is 0 at the dose 2 μg Vip3Aa per cm^2, with the selection doses (all > 2) all or close to all heterozygous larvae would be killed by Vip3Aa in each round of selection. One would expect the resistance level would have increased rather quickly by selection with Vip3Aa, as in theory only homozygous highly resistant larvae would survive the selection. However, after one year of selections with the toxin, the resistance level was only 7.4-fold. The resistance level increased rapidly only after 2021.05. Does this observation suggest the CHS2 gene mutation occurred in the lab in the selection process at one point during that time?

3. In Fig 1, please explain in legend what the solid line indicates (mean of deltaSNP index in 1 kb window?).

4. In line 142 "Results from Sanger sequencing of the full-length genomic sequences of SfCHS2 …": Methods for this experiment are missing the manuscript.

5. The sentence from line 214-216 does not tell much significant, as the important event is mutation of CHS2 causing resistance to Vip3Aa and transposition of a transposon is not likely gene specific for Vip resistance or Cry resistance.

6. The conclusive sentence from line 225 to 227 "The finding … suggests that this domain plays a key role in susceptibility to Vip3Aa": I would suggest remove this sentence, as there is no evidence that the truncated protein remains to be a membrane protein in the midgut as the wild type.

7. Line 247 to 249, authors stated the finding of "Yaoer" from the SS strain "originated in Ruili City". Please provide a citation or the result in this manuscript.

8. In Methods, numerous biocomputing software packages were used, but no references were provided for the software. Please add reference or sources for these software in Methods.

Reviewer #2: 

In this manuscript by Liu and colleagues a compelling case is made that very high levels of resistance to the insecticidal protein VIP3A are generated by mutations in a chitin synthase gene of pest moths. The story starts with 17 generations of lab selection on the fall army worm (Spodoptera frugiperda; a major invasive pest of agriculture) to ever increasing doses of VIP3a, which is a bacterial derived toxin that has a different (but poorly characterised) mode of action than Bt Cry toxins. Genetic mapping shows that gene is autosomal, its dominance depends on the dose, and is located on chromosome 1. Then bulked segregant analysis is used to map the gene at a fine scale and with some sensible elimination of candidates based on their expression in the midgut (where these toxin act) and to proteins that are membrane bound the authors are lead to a chitin synthase thought to have a role in generating the peritrophic membrane (that is sloughed off contributing to bolus during digestion). The putative causal mutation is a Transposable element (they named Yaoer) that happens to have inserted within the splice donor of intron 21. CRISPR experiments in the fall army worm and two other species irrefutably confirm the role of this gene as mutations that must inactivate the gene also give VIP3A resistance. 

This represents a very significant advance in the field as VIP3A resistance has not been characterised to gene-resolution before and the finding will shape the way that field resistance is interpreted and screened for by researchers from now on. 

The experimental logic that unfolds throughout the paper makes it an enjoyable read. I do however have a few comments and questions. 

1. The resistance variant would have been present in the moths that were used to start the lab colony. It did not arise via mutation in the lab colony because the same mutation has been found in 1/540 moths collected independently. If samples had been kept it'd be interesting to retrospectively track its frequency through the 17 generations of selection. Regardless the methods should state the approximate size of the original lab population. 

2. The paragraphs initial describing the transposon would benefit from more attention. In particular it seems to me that 3' Rapid Amplification of cDNA Ends (RACE) would give a more satisfactory understanding of the transcriptional consequences of the insertion - rather than the hypothetical possibilities articulated in FigS3. Perhaps this is what ISO-Seq does - but if so either be more explicit or provide a method citation. And the text about the 5' splice site being downstream from Yaoer is very confusing. Also line 142 refers to Sanger sequencing of a PCR product of at least 6427bp- I don't think the implied cloning is mentioned in the methods. 

3. It is interesting that the Yaoer element does not appear to disrupt the whole transcript but just the end of the transcript (text and figure 2C). This suggests that nonsense mediated decay does not occur. The authors note that the truncated protein may have some residual function. But what of the CRISPR knockouts that occur in exon 5 to 6. Presumably the transcripts produced from that are produced from them are not stable and yet homozygotes are viable. Is the fitness cost of the CRISPR knockouts less than the fitness cost of the Yaoer bearing alleles? (the figure legend of figure 2 does not state whether section c is from gDNA or cDNA nor are the molecular markers used stated). 

4. I note a phylogenetic tree of the chitin synthase gene is provided in the sups. Personally I'd prefer a figure of the enzymatic reaction that the chitin synthase performs. Perhaps that could be added as well? 

5. line 90 refers to non-Bt maize , and by implication this suggests that VIP3Aa is from Bt (even though it is not a Cry protein) which is no doubt obvious to authors but rewording would make it clearer to more generalist readers.

---

## [Editor Report · Decision Letter 2]

29 May 2024

Dear Dr Xiao,

Thank you for your patience while we considered your revised manuscript "Retrotransposon-mediated disruption of a chitin synthase gene confers insect resistance to Bacillus thuringiensis Vip3Aa toxin" for publication as a Short Reports at PLOS Biology. This revised version of your manuscript has been evaluated by the PLOS Biology editors, and the Academic Editor.

Based on our Academic Editor's assessment of your revision, we are likely to accept this manuscript for publication. Please also make sure to address the following data and other policy-related requests.

Please supply the numerical values either in the a supplementary file or as a permanent DOI’d deposition for the following figures:

Figure 1AB, 2EFGHIJ, 3BC, and for Tables S1, S2, S3, S5, S7, S8

b) Please cite the location of the data clearly in all relevant main and supplementary Figure legends, e.g. “The data underlying this Figure can be found in S1 Data” or “The data underlying this Figure can be found in https://doi.org/10.5281/zenodo.XXXXX”

c) Please provide the tree files for Figure S2

d) Please ensure that your Data Statement in the submission system accurately describes where your data can be found and is in final format, as it will be published as written there.

e) Per journal policy, if you have generated any custom code during the curse of this investigation, please make it available without restrictions upon publication. Please ensure that the code is sufficiently well documented and reusable, and that your Data Statement in the Editorial Manager submission system accurately describes where your code can be found.

We expect to receive your revised manuscript within two weeks. 

*Published Peer Review History*

*Press*

Sincerely,

Melissa

Melissa Vazquez Hernandez, Ph.D.

Associate Editor

PLOS Biology

---

## [Editor Report · Decision Letter 3]

7 Jun 2024

Dear Dr Xiao,

Thank you for the submission of your revised Short Reports "Retrotransposon-mediated disruption of a chitin synthase gene confers insect resistance to Bacillus thuringiensis Vip3Aa toxin" for publication in PLOS Biology. On behalf of my colleagues and the Academic Editor, Louis Lambrechts, I am pleased to say that we can in principle accept your manuscript for publication, provided you address any remaining formatting and reporting issues. These will be detailed in an email you should receive within 2-3 business days from our colleagues in the journal operations team; no action is required from you until then. Please note that we will not be able to formally accept your manuscript and schedule it for publication until you have completed any requested changes.

PRESS

Sincerely, 

Melissa

Melissa Vazquez Hernandez, Ph.D., Ph.D.

Associate Editor

PLOS Biology
